# Design, Synthesis, and Biological Evaluation of 2-Mercaptobenzoxazole Derivatives as Potential Multi-Kinase Inhibitors

**DOI:** 10.3390/ph16010097

**Published:** 2023-01-09

**Authors:** Mohammed M. Alanazi, Saleh Aldawas, Nawaf A. Alsaif

**Affiliations:** Department of Pharmaceutical Chemistry, College of Pharmacy, King Saud University, Riyadh 11541, Saudi Arabia

**Keywords:** anticancer, docking, protein kinases, benzoxazole, N-acylhydrazone (NAH), isatins

## Abstract

A series of 12 compounds was designed and synthesized, based on 2-mercaptobenzoxazole derivatives containing either the substituted benzenes **4a**–**d**, substituted isatins **5a**–**f**, or heterocycles **6a**–**b**. The in vitro antiproliferative activity of the compounds was evaluated against hepatocellular carcinoma (HepG2), mammary gland cancer (MCF-7), breast cancer (MDA-MB-231), and the epithelioid cervix carcinoma (HeLa) cancer cell lines. Compounds **4b**, **4d**, **5d**, and **6b** had the most potent antiproliferative activity, with IC_50_ values ranging from 2.14 to 19.34 µM, compared to the reference drugs, doxorubicin and sunitinib. Compound **6b** revealed a remarkably broad antitumor activity pattern against HepG2 (IC_50_ 6.83 µM), MCF-7 (IC_50_ 3.64 µM), MDA-MB-231 (IC_50_ 2.14 µM), and HeLa (IC_50_ 5.18 µM). In addition, compound **6b** showed potent inhibitory activities against EGFR, HER2, VEGFR2, and the CDK2 protein kinase enzymes, with IC_50_ values of 0.279, 0.224, 0.565, and 0.886 µM, respectively. Moreover, compound **6b** induced caspase-dependent apoptosis and cell cycle arrest at the G2/M phase. Finally, a molecular docking simulation was performed for compound **6b** to predict the potential ligand–protein interactions with the active sites of the EGFR, HER2, and VEGFR2 proteins.

## 1. Introduction

Cancer is a very complex disease, compared to other conditions, and has led to a variety of cellular abnormalities and genetic disorders. Worldwide, cancer may be considered the most widespread health apprehension illness that affects human beings. It is known as a group of illnesses that are characterized by unregulated cell growth and division [1]. Globally speaking, cancer is among the leading causes of mortality and morbidity, affecting almost every region at a socioeconomic level [2]. According to the figures released by the World Health Organization (WHO), the prevalence of cancer disease is anticipated to rise by nearly 22 million cases, with 13 million deaths by 2030 [3]. There are two leading causes of abnormal growth in human cancers: the first is the disturbance of the signaling pathways in the cells and the second is the disruption of the cell cycle [4,5,6]. Drug resistance is considered one of the main difficulties in cancer therapy. It may be developed in all types of cancers and in therapeutic strategies such as immunotherapy, molecularly targeted therapy, and chemotherapy [7].

Protein kinase enzymes have a vital role in the signal transduction processes that control many cellular functions. In the last few decades, kinase enzyme inhibitors have received attention because of their current role in treating cancer. The epidermal growth factor receptor (EGFR) is a tyrosine kinase transmembrane receptor that regulates many signal transduction pathways (Ras/MAPK, Jak/STAT, and PI3K/Akt) that control cell growth, division, and apoptosis [8,9]. The overexpression of EGFR is involved in varied types of cancer, such as breast, ovarian, prostate, and colon cancers, by stimulating neoplastic cell proliferation, metastasis, angiogenesis, and invasiveness [8]. Furthermore, the EGFR-signaling cascades activate the vascular endothelial growth factor (VEGF), which is recognized as the main stimulator of angiogenesis. Conversely, the interaction between vascular endothelial growth factor receptor 2 (VEGFR-2) and VEGF will activate signaling cascades (PI3K/Akt and p38MAPK) that affect many cellular functions, such as migration, survival, vascular permeability, and the proliferation of cancer tissue cells, and, hence, trigger angiogenesis [9]. Moreover, it was determined that VEGFR-2 is highly expressed in tumor cells, particularly in epithelial tissue cells. Additionally, each EGFR, VEGFR-2, and glycoprotein is structurally related; the suppression of EGFR reduces the expression of VEGF, while the targeting of VEGFR-2 induces the antitumor effect of the EGFR inhibitors. Hence, current therapeutic strategies support the twin suppression of each VEGFR-2, and EGFR represents a promising cancer-fighting strategy [8].

CDKs are defined as a large family of serine/threonine kinase enzymes that catalyze the phosphorylation of proteins [10]. More than 30 cyclin proteins have been identified in the human body, based on the box domain of the cyclin protein, which can bind and activate the CDKs. The dysfunction and/or overexpression of cyclins or CDKs have been recognized in many human cancers and various pathologies. Thus, these protein kinase enzymes are considered potential therapeutic goals for drug design and development [11,12].

One of the most interesting heterocycles in medicinal chemistry is benzoxazole. Benzoxazole derivatives have drawn much attention in the last few decades, due to their utility as intermediates for synthesizing novel, biologically active molecules. Moreover, benzoxazole derivatives display a wide range of pharmacological activities, comprising anticancer, antiviral, antifungal, and anti-inflammatory activities [13,14]. A benzoxzole derivative (**I**) demonstrated promising antiproliferative activity against various distinct cancer cell lines, with IC_50_ values in the submicromolar range (Figure 1).

Another important nucleus in medicinal chemistry is the N-acylhydrazone. The N-acylhydrazone scaffold (-CO-NH-N=CH-) is illustrated by the fusion between two subunits, imine and amide, which displays a broad spectrum of pharmacological activities, including antibacterial, antihemorrhagic, antitumor, and antidiarrhea properties [15,16,17]. The N-acylhydrazone derivatives are both proton donor and acceptor species, which are a required component of potent kinase inhibitors [18,19,20,21,22]. A derivative of N-acylhydrazone (**II**) was demonstrated to be a potent dual inhibitor of EGFR and VEGFR2 (Figure 1) [23].

Moreover, isatin, or indoline-2,3-dione, is a versatile natural molecule. Isatin and its derivatives have a broad range of pharmacological and biological properties. Isatin derivatives have been promoted as antimicrobial, anticonvulsant, antianxiety, antitumor, antimalarial, antifungal, and antioxidant therapeutics [24,25,26,27,28,29,30]. The success of the isatin scaffold as a new class of antitumor therapeutics is confirmed by the current approval of some of its derivatives, such as sunitinib maleate (**III**) (marketed as Sutent^®^), by the FDA as a chemotherapeutic agent for the treatment of two different types of cancer at the same time: advanced renal carcinoma and gastrointestinal stromal cancers (Figure 1) [31]. Additionally, toceranib phosphate (**IV**) (marketed as Palladia^®^) is an orally bioavailable drug that is a structural analog of sunitinib, showing a potent suppression effect against some of the protein kinase receptor family (Figure 1) [32].

In this study, we sought to develop effective antitumor agents by synthesizing the novel hybrid derivatives, **4a**–**6b**, of 2-mercaptobenzoxazole derivatives conjugated by a hydrazone linker to 5-substituted isatin, substituted benzene, thiazole or pyrrole (Figure 1), which were evaluated for their anticancer activity. Furthermore, to identify the most active derivative among the new compounds, they were tested for their inhibitory activity against the EGFR, HER2, CDK2, and VEGFR2 kinase enzymes and as apoptosis and caspase inducers. Molecular docking simulations were studied to predict the potential binding interactions between the kinase enzymes and the target compounds.

## 2. Results and Discussion

### 2.1. Chemistry

The strategy to prepare the designed 2-mercaptobenzoxazole derivatives is outlined in Figure 1. In the initial step, ethyl 2-(2-mercaptobenzoxazole) acetate (**2**) was synthesized by refluxing 2-mercaptobenzoxazole (**1**) and ethyl chloroacetate in dry acetone as a solvent for 5–10 h, using anhydrous potassium carbonate as a basic catalyst [13]. This was followed by hydrazide formation through the reaction of the resulting ethyl ester (**2**) with hydrazine hydrate in ethanol and reflux for 6–10 h, to yield the hydrazide derivative (**3**). Next, the obtained hydrazide derivative (**3**) was subjected to Schiff‘s base formation through the nucleophilic addition–elimination reaction with the appropriate aldehyde or ketone. Finally, to obtain derivatives **4(a**–**d)**, the hydrazide derivative (**3**) was dissolved in the appropriate amount of ethanol, then mixed with the appropriate benzaldehyde derivative. Acetic acid was then added as a catalyst and the mixture was refluxed for around 6–10 h at 90 °C. The same procedure was applied to synthesize compounds **5(a**–**f)**, but with the replacement of the benzaldehyde derivatives with isatin derivatives. Thiophene-2-carbaldehyde and 1H-pyrrole-2-carbaldehyde were used as the carbonyl part in Schiff‘s base formation for the synthesis of compounds **6(a**–**b)**, according to the procedure mentioned in the experimental section.

The physicochemical characteristics of the newly prepared compounds were determined and reported in the experimental section. The molecular structures of the prepared molecules, **4(a**–**d)**, **5(a**–**f),** and **6(a**–**b)**, were confirmed by spectroscopic analytical methods, including ^1^H-NMR (DMSO-d6, 700 MHz, ppm), ^13^C-NMR (DMSO-d6, 176 MHz, ppm), as well as melting point, elemental analysis, and mass spectral analyses.

In the case of the 1H-NMR spectra, the presence of multiple signals between 7 and 9 ppm reflected the presence of aromatic protons in the prepared derivatives. All the synthesized derivatives, without any exception, have a diastereomeric center at the -N=C- bond; hence, they exist as a E/Z diastereomeric mixture with variable ratios. The synthesized derivatives have a singlet signal at the aliphatic region (~4.25–4.95 ppm) of the methylene -CH2- protons. Compounds **4d**, **5b**, and **5d** have an extra signal at the aliphatic part, due to the presence of the -CH3 group, which appears as a singlet at 3.85, 2.30, and 3.80 ppm, respectively. Compound **4d** showed a broad singlet at 9.57 and 9.59 ppm for the E-diastereomer and Z-diastereomer, respectively, because of the aromatic hydroxyl group. Finally, DMSO-d6 was used for recording the ^13^C-NMR spectra of 2-mercaptobenzoxazole derivatives; it was observed that the spectral signals and proposed molecular structure of the synthesized molecules showed good agreement. The reaction times (h), E:Z ratio, the percentage of the yields, and the melting points of the products are given in Table 1.

### 2.2. Biological Evaluation

#### 2.2.1. Cytotoxicity Assay

The in vitro cytotoxicity for the final target compounds, **4(a**–**d)**, **5(a**–**f)**, and **6(a**–**b)**, was preliminarily evaluated via the MTT colorimetric assay against a panel of four human cell lines, namely, the hepatocellular carcinoma (HepG2), mammary gland (MCF-7), breast cancer (MDA-MB-231) and epithelioid cervix carcinoma (HeLa) [33,34,35,36]. Doxorubicin and sunitinib were used as reference drugs [37,38]. The activities of the tested compounds are expressed as IC_50_ values (µM) and are shown in Table 2. The tested molecules exhibited varying degrees of cytotoxic activities and potencies against the investigated cell lines, HepG2, MCF7, MDA-MB-231, and HeLa cell lines, compared to the positive control reference drugs.

As a general pattern, the breast cancer (MDA-MB-231) cell line was the cancer cell line most sensitive to the synthesized compounds. Compound **4b** displayed moderate anticancer activity against the tested cell lines, with IC_50_ values ranging from 9.72 to 19.34 µM. Moreover, compounds **4d** and **5d** were very potent and show highly promising activity against all the tested cell lines, with IC_50_ values ranging from 2.14 to 12.87 µM. Additionally, compound **6b** revealed similar inhibitory activity with the reference drugs doxorubicin and sunitinib, against the HepG2, MCF-7, MDA-MB-231, and HeLa cell lines, with IC_50_ values of 6.83, 3.64, 2.14, and 5.18, respectively. On the other hand, the remaining compounds demonstrated modest antiproliferative activities against the tested cell lines.

#### 2.2.2. Structure–Activity Relationship (SAR)

In series 4 and 5, compounds with a methoxy substitution (compounds **4d** and **5d**) were the most active compounds against the four cell lines. Replacing the substituted phenyl in series 4 or the substituted isatin in series 5 with a pyrrole ring system (compound **6b**) resulted in a significant increase in potency against the four cell lines, with slightly better potency than the potency of the reference compound sunitinib, especially against the MDA-MB-231 cell line.

#### 2.2.3. In Vitro Kinase Inhibitory Activity

Compound **6b**, due to its excellent cytotoxic effect, was also subjected to further biological investigations, including enzyme assays against CDK2, EGFR, HER2 and the VEGFR2 protein kinases, in a trial to confirm the mechanism of its cytotoxicity.

As shown in Table 3, the results revealed that compound **6b** has good inhibitory activity in the nanomolar range on the CDK2, EGFR, HER2, and VEGFR2 protein kinases compared to the reference drugs, with HER2 displaying the most sensitivity, followed by EGFR, CDK2, and VEGFR2. The average IC_50_ values of the respective kinases were 0.224, 0.279, 0.886, and 0.565 μM, respectively. These results confirm the previous speculation that these molecules may have multiple cellular targets, including the inhibition of these kinase enzymes, and this may aid in overcoming the drug resistance phenomenon for many of the clinically employed anticancer agents. Consequently, more biological studies should be performed to explore the mechanism of action inside cancer cells.

#### 2.2.4. Molecular Docking

To rationalize the cytotoxic activities and predict the possible types of drug–receptor interactions of the synthesized compounds, the most potent compound, **6b**, was docked into the active binding sites of EGFR, VEGVR2, and HER2. The co-crystalized ligands, erlotinib, sorafenib, and lapatinib, were used as reference compounds for EGFR, VEGVR2, and HER2, respectively. First, erlotinib and compound **6b** were docked into the ATP-binding site of EGFR. Erlotinib created one hydrogen bond with Met769 and multiple hydrophobic interactions with Lys721, Val702, Ala719, Leu820, Leu694, Thr830, and Met769. Conversely, compound **6b** formed two hydrogen bonds with Asp831 and Thr830 and several hydrophobic interactions with Leu694, Ala719, Val702, Leu820, Met742, Leu742, Leu753, and Cys751. It is obvious that compound **6b** forms different hydrogen bonds than erlotinib, while the hydrophobic interactions were almost the same; however, compound **6b** was superimposed on erlotinib in the active site of EGFR (Figure 2 and Appendix A). The binding energies of erlotinib and compound **6b** with the EGFR were −9.5 and −8.0 Kcal/mol, respectively. Second, docking sorafenib into the active site of VEGFR2 resulted in four hydrogen bonds with Cys919 (two hydrogen bonds), Asp1046, and Glu885, and several hydrophobic interactions with the amino acid residues of the active pocket of VEGFR2. The oxygen atom of the benzoxazole ring of compound **6b** served as a hydrogen bond acceptor by making a bond with Asp1046, while other functional groups stabilized the compound by making hydrophobic interactions with Leu889, Ala866, Leu1035, Cys919, Leu840, and Val848. In comparing compound **6b** with sorafenib, compound **6b** was superimposed onto sorafenib in the active site of VEGFR2; however, it then made fewer hydrogen bonds and hydrophobic interactions (Figure 3 and Appendix A). Sorafenib and compound **6b** showed binding affinities of −11.9 and −8.4 Kcal/mol, respectively, with VEGFR2. Lastly, lapatinib and compound **6b** were docked into the active binding site of the HER2 receptor. Lapatinib superimposed lapatinib and interacted with the amino acid residues of Met801, Thr798, and Thr862 by hydrogen bonds, with Ser783 and Arg784 by halogen bonds, and with Arg784, Leu785, Leu796, Val734, and Lys753 by hydrophobic interaction. Compound **6b** formed two hydrogen bonds with Lys753 and The862, in addition to several hydrophobic interactions with Ala751, Leu796, Thr862, val734, and Leu 852 (Figure 4 and Appendix A). The binding energy with HER2 was found to be −9.8 and −8.7 Kcal/mol for lapatinib and compound **6b**, respectively.

#### 2.2.5. Bax, Bcl-2, Caspase-3, and Caspase-9 Gene Expression Screening

Compound **6b** was selected to investigate the apoptotic markers against the liver cancer cell line (HepG2) [34]. It exhibited good cytotoxicity toward the tested cell lines, with significantly low IC_50_ values ranging from 2 to 6 μM. Typically, Bax, Bcl-2, and caspases contribute to the regulation of apoptotic signaling. Acting as an apoptotic activator (Bax) or inhibitor (Bcl-2), the Bcl-2 family of proteins plays a significant role in apoptosis [39]. The caspase family comprises cysteine proteases that are classified as either executioner—caspase-3, or as initiators—caspase-9 and caspase-8 [40]. The results in Table 4 showed the strong stimulated expression of the pro-apoptotic Bax gene (5.34 folds) and apoptotic genes, along with Caspase-3 (5.02 folds) and Caspase-9 (3.5 folds), compared to the negative controls. Nevertheless, considerable downregulation of the anti-apoptotic gene, Bcl-2 (0.26 folds), was described, leading to an increased expression ratio of Bax/Bcl-2 (1→20.53). The Bax/Bcl-2 gene expression ratio can serve as an early predictor for cancer in patients and is a sensitive monitor of cancer progression [41]. Collating all the provided evidence, the upregulation of caspase-3, caspase-9, and Bax is clear, while the downregulation of Bcl-2 genes makes it evident that compound **6b** induces apoptosis within the liver cancer cell line (HepG2).

#### 2.2.6. Apoptosis Rate and Cell-Cycle Analysis

Investigation of the mechanistic growth-inhibitory action of compound **6b** on liver cancer cells (HepG2), a cell-cycle and apoptosis rate analysis was conducted at the compound’s approximated IC_50_ value (6.83 μM). Programmed cell death is a fundamental cellular program that is inherent in every cell of the human body. One of the most extensively studied forms of programmed cell death is apoptosis, which plays an important role during the various physiological processes and is involved in a variety of pathological conditions [42]. The HepG2 cells were exposed to compound **6b** for 24 h and the cell cycle was monitored via flow cytometry; the results are reported in Table 5.

The impact of compound **6b** on the cell cycle distribution revealed that the predominant cell population at the G2/M stage (31.26%) was significantly higher than that of the untreated cell line (7.12%). Furthermore, a significant reduction at the S stage complemented the elevated cell population at the G2/M stage of the treated cells compared to the negative controls (28.37% vs. 43.16%) (see Figure 5). 

To identify the mode of cell death promoted by compound **6b** within the liver cancer cells (HepG2), apoptosis rate analysis was performed, following 24 h of exposure. Compound **6b**, at a concentration of 6.83 μM, induced both early- and late-stage apoptosis in the HepG2 cell line, with significantly elevated percentage apoptotic cell levels compared to the controls (1.64% and 10.91%, respectively). Moreover, the average proportion of Annexin-V-stained positive cells (total apoptotic cells) was elevated from 2.02% within the untreated cells to 16.84% within the treated ones. Interestingly, compound **6b** showed little influence on the necrosis of liver cancer cells. The provided findings are supported by the previous cell cycle analysis, confirming the potentiality of **6b** as a promising anticancer agent (see Table 6 and Figure 6).

## 3. Materials and Methods

### 3.1. Chemistry and Analysis

Melting points were recorded in open capillaries, using an electrothermal 9200 melting point apparatus (Cole-Parmer GmbH, Wertheim, Germany). Nuclear magnetic resonance (NMR) spectra were recorded (700 MHz for proton ^1^H and 176 MHz for carbon ^13^C) in deuterated dimethyl sulfoxide (DMSO-d6) as a solvent and tetramethylsilane (TMS) as an internal standard, using Bruker spectrometers (Bruker, Coventry, Germany) in the College of Pharmacy, King Saud University, Saudi Arabia. All chemical shifts were expressed per the δ scale (ppm), while coupling constants (*J*) for ^1^H were given in Hz and expressed as (s) for a singlet, (bs) for a broad singlet, (d) for a doublet, (t) for a triplet, (dd) for a doublet of doublets, (td) for a triplet of doublets, and (m) for a multiplet. Mass spectra were recorded using an Agilent Single Quad mass spectrometer (Bruker, Billerica, MA, USA). Elemental analyses were performed on a Perkin Elmer 2400 CHN elemental analyzer (PerkinElmer, Inc., Waltham, MA, USA). Reactions were monitored via thin-layer chromatography (TLC) using systems (ethylacetate 20%, hexane 80%) and (methanol 5%, chloroform 95%), while spots were visualized using an ultraviolet lamp.

#### 3.1.1. General Protocol for the Synthesis of Ethyl 2-[(1,3-benzoxazol-2-yl)sulfanyl]acetate (**2**) 

An appropriate volume of ethyl chloroacetate (0.023 mol) was added to a stirred solution of 2-mercaptobenzoxazole, compound **1** (0.02 mol), and anhydrous potassium carbonate (0.02 mol) in 30 mL of acetone, then refluxed for 5–10 h. After that, the reaction mixture was cooled, and the solution was evaporated under a reduced pressure to remove the solvent. The pure oily product was then obtained [43,44,45].

#### 3.1.2. General Protocol for the Synthesis of 2-[(1,3-benzoxazol-2-yl) sulfanyl] Acetohydrazide (**3**)

Compound **2** (0.01 mol) and hydrazine hydrate (0.05 mol) were mixed well and heated over a water bath for 10 min before adding 30 mL of ethanol. The reaction mixture was then heated with reflux for 6–10 h. When the reaction was completed, the mixture was cooled to room temperature, added to iced water, filtered, and, finally, recrystallized from the absolute ethanol [43,44,45]. 

#### 3.1.3. General Protocol for the Synthesis of (E/Z)-N′-(substituted benzylidene)-2-(benzo[d]oxazol-2-ylthio)acetohydrazide (4**(a**–**d)**), (E/Z)-2-(benzo[d]oxazol-2-ylthio)-N′-(5-substituted/unsubstituted-2-oxoindolin-3-ylidene)acetohydrazide (**5(a**–**f)**) and (E/Z)-2-(benzo[d]oxazol-2-ylthio)-N′-((thiophen-2-yl)/(1H-pyrrol-2-yl)methylene)acetohydrazide **6(a**–**b)**]

A series of compounds **4(a**–**d)**, **5(a**–**f)**, and **6(a**–**b)** were synthesized by dissolving carbohydrazides **3** (1.3 mmol) and substituted benzaldehyde (1.3 mmol), heterocyclic aldehyde (1.3 mmol), or isatin (1.3 mmol) in 20 mL of ethanol and two drops of acetic acid. The reaction mixture was stirred for 3–6 h at 80 °C. The progress of the reaction was monitored by TLC. After completion, the reaction mixture was poured over ice-cold water, then neutralized with ammonia. The precipitated product was filtered, washed with water, and recrystallized from a mixture of absolute methanol and water (95%:5%, respectively) to yield the desired product [44,46].

##### Synthesis of (E/Z)-N′-(3,4-dichlorobenzylidene)-2-(benzo[d]oxazol-2-ylthio)acetohydrazide (**4a**)

Yield 86%; m.p. 193–195 °C; ^1^H-NMR (DMSO-d6, 700 MHz) 12.05 (bs, 0.3H, NH), 11.94 (bs, 0.7H, NH), 8.22 (s, 0.3H, =CH-), 8.04 (s, 0.7H, =CH-), 8.01 (s, 0.7H, Ar-H), 7.96 (s, 0.3H, Ar-H), 7.76–7.70 (m, 2H, Ar-H), 7.69–7.62 (m, 2H, Ar-H), 7.38–7.31 (m, 2H, Ar-H), 4.71 (s, 1.4H, -CH_2_-), 4.30 (s, 0.6H, -CH_2_-).; ^13^C-NMR (DMSO-d6, 176 MHz) δ168.75, 164.15, 164.47, 163.64, 151.82, 151.79, 145.07, 141.94, 141.71, 141.64, 135.37, 135.26, 132.84, 132.67, 132.22, 132.14, 131.58, 131.54, 129.26, 128.97, 127.34, 127.31, 125.18, 125.12, 124.87, 124.76, 118.75, 118.74, 110.75, 110.68, 35.20, 35.03; MS (*m*/*z*): 380; elemental analysis for C_16_H_11_Cl_2_N_3_O_2_S (380.25) calc./found; C, 50.54/50.75; H, 2.92/3.21; N, 11.05/11.28.



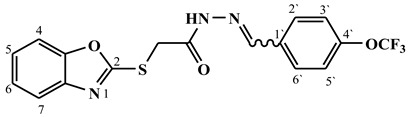



##### Synthesis of (E/Z)-N′-(4-(trifluoromethoxy)benzylidene)-2-(benzo[d]oxazol-2-ylthio)acetohydrazide (**4b**)

Yield 81%; m.p. 154–156 °C; ^1^H-NMR (DMSO-d6, 700 MHz) 11.96 (bs, 0.3H, NH), 11.87 (bs, 0.7H, NH), 8.27 (s, 0.3H, =CH-), 8.09 (s, 0.7H, =CH-), 7.87 (d, *J* = 7.5 Hz, 2H, Ar-H), 7.69–7.62 (m, 2H, Ar-H), 7.45 (d, *J* = 7.2 Hz, 2H, Ar-H), 7.38–7.31 (m, 2H, Ar-H), 4.70 (s, 1.4H, -CH_2_-), 4.30 (s, 0.6H, -CH_2_-).;^13^C-NMR (DMSO-d6, 176 MHz) δ168.63, 164.49, 164.19, 163.50, 151.82, 151.79, 149.68, 146.16, 142.98, 141.71, 141.65, 133.78, 133.70, 131.03, 129.53 (2C), 129.31 (2C), 125.18, 125.13, 124.87, 124.76, 121.83 (2C), 121.77 (2C), 121.22, 119.76, 118.74, 118.73, 110.74, 110.69, 35.23, 35.07.; MS (*m*/*z*): 396.10; elemental analysis for C_17_H_12_F_3_N_3_O_3_S (395.36) Calc./Found; C, 51.65/51.79; H, 3.06/3.17; N, 10.63/10.85.



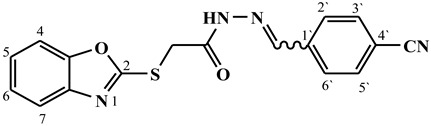



##### Synthesis of (E/Z)-N′-(4-cyanobenzylidene)-2-(benzo[d]oxazol-2-ylthio)acetohydrazide (**4c**)

Yield 74%; m.p. 246–248 °C; ^1^H-NMR (DMSO-d6, 700 MHz) 12.10 (bs, 0.29H, NH), 12.01 (bs, 0.71H, NH), 8.30 (s, 0.29H, =CH-), 8.11 (s, 0.71H, =CH-), 7.95–7.89 (m, 4H, Ar-H), 7.69–7.62 (m, 2H, Ar-H), 7.38–7.31 (m, 2H, Ar-H), 4.72 (s, 1.42H, -CH_2_-), 4.31 (s, 0.58H, -CH_2_-).; ^13^C-NMR (DMSO-d6, 176 MHz) δ168.88, 164.43, 164.14, 163.76, 151.83, 151.79, 145.81, 142.64, 141.70, 141.64, 138.99, 138.84, 133.47, 133.22 (2C), 133.21 (2C), 128.20 (2C), 128.03 (2C), 125.18, 125.14, 124.87, 124.78, 119.14, 119.10, 118.75, 112.49, 112.33, 110.69, 110.75, 35.24, 35.00; MS (*m*/*z*): 337.10; elemental analysis for C_17_H_12_N_4_O_2_S (336.37) Calc./Found; C, 60.70/60.54; H, 3.60/3.81; N, 16.66/16.94.



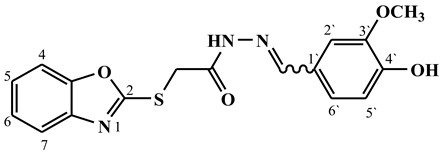



##### Synthesis of (E/Z)-N′-(4-hydroxy-3-methoxybenzylidene)-2-(benzo[d]oxazol-2-ylthio)acetohydrazide (**4d**)

Yield 60%; m.p. 150–153 °C; ^1^H-NMR (DMSO-d6, 700 MHz) 11.68 (bs, 0.36H, NH), 11.61 (bs, 0.64H, NH), 9.59 (bs, 0.36H, OH), 9.57 (bs, 0.64H, OH), 8.10 (s, 0.36H, =CH-), 7.94 (s, 0.64H, =CH-), 7.70–7.61 (m, 2H, Ar-H), 7.38–7.26 (m, 3H, Ar-H), 7.12 (d, *J* = 8.0 Hz, 0.64H, Ar-H), 7.10 (d, *J* = 8.0 Hz, 0.36H, Ar-H), 6.87–6.83 (m, 1H, Ar-H), 4.67 (s, 1.28H, -CH_2_-), 4.26 (s, 0.72H, -CH_2_-), 3.85 (s, 1.92H, -CH_3_), 3.83 (s, 1.08H, -CH_3_).; ^13^C-NMR (DMSO-d6, 176 MHz) δ168.12, 168.02, 164.67, 164.22, 151.77, 151.59, 149.55, 149.34, 148.42, 148.25, 144.93, 141.72, 141.66, 136.04, 125.82, 125.75, 125.18, 125.12, 124.87, 124.74, 122.65, 121.77, 118.74, 118.70, 116.03, 115.88, 110.73, 110.66, 110.19, 109.59, 56.51, 56.04, 35.21, 35.10; MS (*m*/*z*): 355.90; elemental analysis for C_17_H_15_N_3_O_4_S (357.38) Calc./Found; C, 57.13/57.30; H, 4.23/4.49; N, 11.76/11.92.



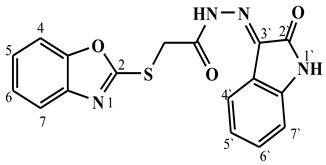



##### Synthesis of (E/Z)-2-(benzo[d]oxazol-2-ylthio)-N′-(2-oxoindolin-3-ylidene)acetohydrazide (**5a**)

Yield 75%; m.p. 223–225 °C; ^1^H-NMR (DMSO-d6, 700 MHz) 11.66 (bs, 0.7H, NH-Hydrazide), 11.48 (bs, 0.3H, NH-Hydrazide), 10.87 (bs, 1H, NH-Indole), 8.17 (d, *J* = 7.3 Hz, 1H, Ar-H), 7.68–7.65 (d, *J* = 7.0, 1H, Ar-H), 7.65–7.58 (m, 1H, Ar-H), 7.47–7.38 (m, 1H, Ar-H), 7.37–7.30 (m, 2H, Ar-H), 7.17–7.03 (m, 1H, Ar-H), 6.92 (d, *J* = 7.6 Hz, 1H, Ar-H), 4.82 (s, 1.4H, -CH_2_-), 4.58 (s, 0.6H, -CH_2_-).; ^13^C-NMR (DMSO-d6, 176 MHz) δ169.49, 169.18, 165.34, 165.00, 164.35, 164.31, 152.59, 151.85, 151.83, 145.72, 144.56, 144.41, 141.82, 141.64, 139.65, 139.63, 133.27, 126.98, 125.23, 125.19, 124.89, 124.81, 122.30, 122.24, 118.80, 118.73, 115.64, 115.62, 111.22, 111.16, 110.84, 110.75, 35.89, 35.86; MS (*m*/*z*): 375.00; elemental analysis for C_17_H_12_N_4_O_3_S (352.37) Calc./Found; C, 57.95/58.12; H, 3.43/3.60; N, 15.90/16.16.



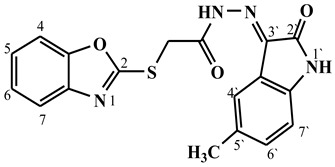



##### Synthesis of (E/Z)-2-(benzo[d]oxazol-2-ylthio)-N′-(5-methyl-2-oxoindolin-3-ylidene)acetohydrazide (**5b**)

Yield 64%; m.p. 230–233 °C; ^1^H-NMR (DMSO-d6, 700 MHz) 13.47 (bs, 0.38H, NH-Hydrazide), 12.76 (bs, 0.62H, NH-Hydrazide), 11.22 (bs, 0.62H, NH-Indole), 11.15 (bs, 0.38H, NH-Indole), 7.67 (d, *J* = 6.7 Hz, 1H, Ar-H), 7.66 (d, *J* = 6.5 Hz, 1H, Ar-H), 7.43–7.30 (m, 3H, Ar-H), 7.22 (m, 1H, Ar-H), 6.86 (d, *J* = 6.7 Hz, 0.38H, Ar-H), 6.82 (d, *J* = 6.7 Hz, 0.62H, Ar-H), 4.83 (s, 1.24H, -CH_2_-), 4.41 (s, 0.76H, -CH_2_-), 2.3 (s, 3H, -CH_3_).; ^13^C-NMR (DMSO-d6, 176 MHz) δ169.63, 169.30, 164.98, 164.11, 163.15, 163.10, 151.86, 151.82, 141.66, 141.63, 140.87, 138.58, 135.80, 132.81, 132.79, 132.25 (2C), 125.20, 125.17, 124.95, 124.88, 121.84, 121.64, 120, 118.94, 118.81, 118.72, 115.70, 111.51, 111.40, 110.78, 110.73, 34.96, 34.26, 20.96, 20.91; MS (*m*/*z*): 367.10; elemental analysis for C_18_H_14_N_4_O_3_S (366.39) Calc./Found; C, 59.01/59.23; H, 3.85/3.77; N, 15.29/15.51.



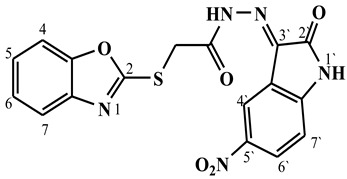



##### Synthesis of (E/Z)-2-(benzo[d]oxazol-2-ylthio)-N′-(5-nitro-2-oxoindolin-3-ylidene)acetohydrazide (**5c**)

Yield 82%; m.p. 165–167 °C; ^1^H-NMR (DMSO-d6, 700 MHz) 12.56 (bs, 0.37H, NH-Hydrazide), 12.40 (bs, 0.63H, NH-Hydrazide), 11.92 (bs, 0.37H, NH-Indole), 11.56 (bs, 0.63H, NH-Indole), 9.16 (s, 0.63H, Ar-H), 8.48–8.28 (m, 1.37H, Ar-H), 7.70–7.60 (m, 2H, Ar-H), 7.37–7.30 (m, 2H, Ar-H), 7.18–7.10 (m, 1H, Ar-H), 4.91 (s, 0.74H, -CH_2_-), 4.83 (s, 1.26H, -CH_2_-).; ^13^C-NMR (DMSO-d6, 176 MHz) δ169.66, 169.62, 165.28, 165.28, 163.95, 163.32, 151.88, 151.83, 149.84, 148.26, 148.24, 143.29, 142.56, 141.65, 141.59, 129.05, 128.09, 125.20, 125.17, 124.91, 124.83, 122.23, 121.45, 120.94, 118.84, 118.74, 116.55, 115.49, 111.92, 111.21, 110.78, 110.73, 35.82, 34.21; MS (*m*/*z*): 398.10; elemental analysis for C_18_H_14_N_4_O_3_S (397.36) Calc./Found; C, 51.38/51.62; H, 2.79/2.95; N, 17.62/17.89.



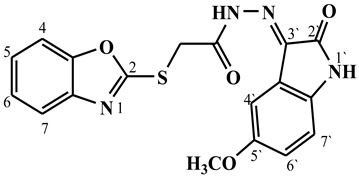



##### Synthesis of (E/Z)-2-(benzo[d]oxazol-2-ylthio)-N′-(5-methoxy-2-oxoindolin-3-ylidene)acetohydrazide (**5d**)

Yield 71%; m.p. 145–147 °C; ^1^H-NMR (DMSO-d6, 700 MHz) 13.52 (bs, 0.42H, NH-Hydrazide), 12.79 (bs, 0.58H, NH-Hydrazide), 11.15 (bs, 0.58H, NH-Indole), 11.08 (bs, 0.42H, NH-Indole), 7.83 (s, 0.58H, Ar-H), 7.70–7.61 (m, 2H, Ar-H), 7.37–7.30 (m, 2H, Ar-H), 7.20 (s, 0.42H, Ar-H), 7.11–6.95 (m, 1H, Ar-H), 6.92–6.82 (m, 1H, Ar-H), 4.85 (s, 1.16H, -CH_2_-), 4.42 (s, 0.84H, -CH_2_-), 3.80 (s, 1.26H, -OCH_3_), 3.78 (s, 1.74H, -OCH_3_).; ^13^C-NMR (DMSO-d6, 176 MHz) δ169.46, 169.34, 165.03, 164.15, 163.15, 162.89, 155.90, 155.89, 151.87, 151.82, 141.70, 141.62, 136.78, 135.92 (2C), 125.20, 125.16, 124.96, 124.87, 120.77, 118.94, 118.80, 118.72, 118.53, 112.57, 112.53, 111.65, 110.84, 110.79, 110.76, 106.45, 106.25, 56.17, 56.09, 34.94, 34.24; MS (*m*/*z*): 383.10; elemental analysis for C_18_H_14_N_4_O_4_S (382.39) Calc./Found; C, 56.54/56.77; H, 3.69/3.84; N, 14.65/14.81.



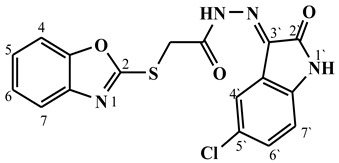



##### Synthesis of (E/Z)-2-(benzo[d]oxazol-2-ylthio)-N′-(5-chloro-2-oxoindolin-3-ylidene)acetohydrazide (**5e**)

Yield 83%; m.p. 264–267 °C; ^1^H-NMR (DMSO-d6, 700 MHz) 13.44 (bs, 0.33H, NH-Hydrazide), 12.65 (bs, 0.67H, NH-Hydrazide), 11.42 (bs, 0.67H, NH-Indole), 11.38 (bs, 0.33H, NH-Indole), 7.70–7.61 (m, 3H, Ar-H), 7.48–7.41 (m, 1H, Ar-H), 7.38–7.31 (m, 2H, Ar-H), 7.02–6.92 (m, 1H, Ar-H), 4.85 (s, 1.34H, -CH_2_-), 4.43 (s, 0.66H, -CH_2_-).; ^13^C-NMR (DMSO-d6, 176 MHz) δ169.58, 169.51, 164.05, 162.79, 162.65, 162.55, 151.97, 151.88, 141.79, 141.60 (2C), 134.61 (2C), 131.75, 131.68, 127.25, 125.26 (2C), 125.07, 125.05, 122.02, 121.84, 121.82, 120.99 (2C), 118.92, 118.82 (2C), 113.22, 113.17, 110.81, 110.78, 34.93, 34.13. MS (*m*/*z*): 385.80; elemental analysis for C_17_H_11_ClN_4_O_3_S (386.81) Calc./Found; C, 52.79/52.95; H, 2.87/3.01; N, 14.48/14.70.



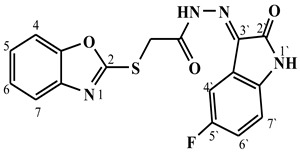



##### Synthesis of (E/Z)-2-(benzo[d]oxazol-2-ylthio)-N′-(5-fluoro-2-oxoindolin-3-ylidene)acetohydrazide (**5f**)

Yield 70%; m.p. 225–227 °C; ^1^H-NMR (DMSO-d6, 700 MHz) 11.80 (bs, 0.67H, NH-Hydrazide), 11.51 (bs, 0.33H, NH-Hydrazide), 10.87 (bs, 1H, NH-Indole), 8.19 (d, *J* = 7.3 Hz, 1H, Ar-H), 7.67 (d, *J* = 6.8 Hz, 1H, Ar-H), 7.64 (m, 1H, Ar-H), 7.38–7.31 (m, 2H, Ar-H), 7.30–7.25 (m, 1H, Ar-H), 6.95–6.90 (m, 1H, Ar-H), 4.85 (s, 1.34H, -CH_2_-), 4.43 (s, 0.66H, -CH_2_-).; ^13^C-NMR (DMSO-d6, 176 MHz) δ168.94, 168.87, 165.20, 165.06, 164.35, 164.26, 158.73, 158.68, 157.35, 156.51, 151.87, 151.82, 141.64, 140.75, 136.46, 136.42, 125.24, 125.18, 124.86, 124.85, 119.32, 119.29, 118.73, 118.71, 116.00, 115.96, 114.09, 113.98, 111.95, 111.92, 110.78, 110.74, 35.85, 35.78; MS (*m*/*z*): 371.10; elemental analysis for C_17_H_11_FN_4_O_3_S (370.36) Calc./Found; C, 55.13/55.34; H, 2.99/3.20; N, 15.13/15.32.



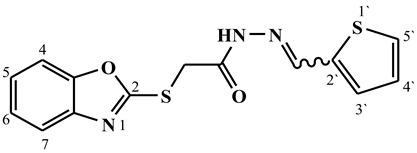



##### Synthesis of (E/Z)-2-(benzo[d]oxazol-2-ylthio)-N′-((thiophen-2-yl)methylene)acetohydrazide (**6a**)

Yield 65%; m.p. 218–221 °C; ^1^H-NMR (DMSO-d6, 700 MHz) 11.82 (bs, 0.38H, NH), 11.76 (bs, 0.62H, NH), 8.45 (s, 0.38H, =CH-), 8.25 (s, 0.62H, =CH-), 7.70–7.62 (m, 3H, Ar-H), 7.49 (d, *J* = 7.3 Hz, 1H, Ar-H), 7.39–7.31 (m, 2H, Ar-H), 7.15 (t, *J* = 6.8, 1H, Ar-H), 4.61 (s, 1.24H, -CH_2_-), 4.26 (s, 0.66H, -CH_2_-).; ^13^C-NMR (DMSO-d6, 176 MHz) δ168.08, 164.55, 164.18, 160.82, 151.82, 151.78, 148.70, 142.90, 141.72, 141.66, 139.74, 138.99, 135.78, 131.80, 131.33, 129.63, 129.24, 128.45, 128.35, 125.18, 125.11, 124.86, 124.74, 118.74, 110.74, 110.67, 35.21, 35.06; MS (*m*/*z*): 318.10; elemental analysis for C_14_H_11_N_3_O_2_S_2_ (317.39) Calc./Found; C, 52.98/53.24; H, 3.49/3.71; N, 13.24/13.50.



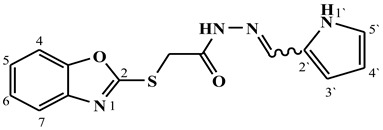



##### Synthesis of (E/Z)-N′-((1H-pyrrol-2-yl)methylene)-2-(benzo[d]oxazol-2-ylthio)acetohydrazide (**6b**)

Yield 67%; m.p. 207–210 °C; ^1^H-NMR (DMSO-d6, 700 MHz) 11.52 (bs, 0.66H, NH), 11.51 (bs, 0.34H, NH), 11.46 (bs, 0.66H, NH-Pyrrole), 11.43 (bs, 0.34H, NH-Pyrrole), 8.07 (s, 0.34H, =CH-), 7.89 (s, 0.66H, =CH-), 7.70–7.62 (m, 2H, Ar-H), 7.39–7.31 (m, 2H, Ar-H), 6.98–6.96 (m, 0.66H, Ar-H), 6.92–6.90 (m, 0.34H, Ar-H), 6.51–6.49 (m, 0.34H, Ar-H), 6.48–6.46 (m, 0.66H, Ar-H), 6.16–6.14 (m, 1H, Ar-H), 4.61 (s, 1.32H, -CH_2_-), 4.26 (s, 0.68H, -CH_2_-).; ^13^C-NMR (DMSO-d6, 176 MHz) δ 167.80, 164.66, 164.29, 162.51, 151.80, 151.76, 141.77, 141.68, 140.83, 137.17, 127.44, 127.13, 125.17, 125.10, 124.83, 124.71, 123.13, 122.57, 118.73, 118.70, 114.08, 113.21, 110.72, 110.65, 109.76, 109.71, 35.36, 35.05; MS (*m*/*z*): 301.10; elemental analysis for C_14_H_12_N_4_O_2_S (300.34) Calc./Found; C, 55.99/56.15; H, 4.03/4.19; N, 18.65/18.57.

### 3.2. Biological Evaluation

#### 3.2.1. Cytotoxicity Assay

The cytotoxicity of the target hybrids was examined by an MTT assay against HepG2, HeLa, MCF-7, and MDA-MB-231. At first, the RPMI-1640 medium with 10% fetal bovine serum was used to cultivate the cell lines. Then, antibiotics (penicillin (100 units/mL) and streptomycin (100 µg/mL)) were added in a 37 °C incubator in 5% CO_2_. The cell lines were seeded in a 96-well plate at a density of 1.0 × 10^4^ cells/well at 37 °C under 5% CO_2_ for 48 h. After incubation, the cells were treated with various concentrations of the synthesized compounds and incubated for an additional 24 h. Following the drug treatment, 20 µL of 5 mg/mL MTT solution was added and incubated for 4 h. To dissolve the purple formazan that had formed, 100 µL dimethyl sulfoxide (DMSO) was added to each well. At a wavelength of 570 nm, the colorimetric assay was recorded and measured, using a plate reader BioTek EXL 800 (Agilent Technologies, Inc., Santa Clara, CA, USA). The percentage of relative cell viability was counted using the following formula (A570 of handled specimens/A570 of the unhandled specimen), multiplied by 100. The results of the cytotoxicity value were presented as the median growth inhibitory concentration (IC_50_) of the reagents compared to the control [47].

#### 3.2.2. In Vitro CDK2, EGFR, HER2, and VEGFR-2 Enzyme Assays

Compound **6b** was biologically evaluated for its inhibitory effect against CDK2, EGFR, HER2, and VEGFR-2. Human CDK2, EGFR, HER2, and VEGFR-2 ELISA kits (Enzyme-Linked-Immunosorbent Serologic Assay) were used during this evaluation. At first, specified antibody proteins were added separately to a 96-well plate, then 0.1 mL of the standard solution or the evaluated molecule was added; these were then incubated for 2.5 h at RT. After rinsing, 100 μL of the ready-made biotin antibody protein was added, then incubated for one extra hour at room temperature and, finally, washed. Next, 100 μL of streptavidin solution was added, incubated at room temperature for 45 min, and then washed. Next, 100 μL of the TMB substrate reagent was added and incubated for 0.5 h at RT, followed by adding 50 μL of the stop solution. The absorbance of the ELISA plate was promptly measured at 450 nm using the ELISA plate reader. The standard curve that was generated had absorbances on the Y-axis and concentrations on the X-axis.

#### 3.2.3. Determination of Apoptosis-Related Proteins

The HepG2 cells were incubated in 96-well plates in triplicates and left for 24 h. After 24 h of incubation, the cells were treated with compound **6b** at different concentrations, while control cells were only treated with 0.1% DMSO (*v*/*v*). The treated cells, along with control cells, were then incubated for another 24 h; then, caspases-3 and 9, BAX, and Bcl-2 levels were determined using ELISA assay kits; KHO1091 (Invitrogen^TM^, Grand Island, NY, USA), EIA-4860 (DRU International, Inc., Mountainside, NJ, USA), EIA-4487 (DRU International INC., Mountainside, NJ, USA) and 99–0042 (Invitrogen^TM^, Grand Island, NY, USA), respectively, according to their manufacturers’ procedures.

#### 3.2.4. Cell-Cycle Flow Cytometry Analysis

To detect the effect of the synthesized compound, **6b**, on cell cycle distribution, flow cytometry analysis was carried out, utilizing propidium iodide (PI) staining. The kit used for the cell cycle flow cytometry analysis was the K101-100 Annexin V-FITC Apoptosis Detection Kit (BioVision, Mountain View, CA, USA). First, HepG2 cells were manipulated with compound **6b** (6.4 µM) for nearly 24 h. Then, the treated cells were fixed in 70% ethanol for 12 h at 4 °C. Following that, the cells were rinsed with cold PBS, incubated with 100 μL RNase A for 0.5 h at 37 °C, and stained with propidium iodide (400 μL) in the dark at RT for an extra 0.5 h. The stained cells were determined, utilizing Epics XLMCL™ flow cytometer equipment (Beckman Coulter, Apeldoorn, Netherlands), then the results were collected and analyzed with the Flowing software (version 2.5.1, Turku Center for Biotechnology, Turku, Finland).

#### 3.2.5. Annexin V-FITC Dual-Staining Apoptosis Analysis

Apoptosis analysis was employed to evaluate the apoptotic effect of compound **6b**. HepG2 cells (2.0 × 10^5^ cells) were treated with compound **6b** (6.4 µM) for 24 h, collected by trypsin, centrifuged, rinsed twice successively with PBS, suspended in 0.5 mL of binding buffer, then dual-stained with Annexin V-FITC (5 μL) and propidium iodide (5 μL) in the dark for 15 min at RT. These stained cells were evaluated using flow cytometry equipment Epics XL-MCL™ (Beckman Coulter, Apeldoorn, Netherlands) with an excitation wavelength of 488 nm and an emission wavelength of 530 nm. The results were then analyzed with the Flowing software.

## 4. Conclusions

A series of 2-mercaptobenzoxazoles, hybridized with either substituted benzenes, substituted isatins, or heterocycles were synthesized and obtained as E/Z-diastereomers. The compounds have been assigned Z or E configurations by 1HNMR analysis. The biological investigations of the synthesized compounds revealed good cytotoxic activities for four compounds against four cancer cell lines: hepatocellular carcinoma (HepG2), mammary gland (MCF-7), breast cancer (MDA-MB-231), and epithelioid cervix carcinoma (HeLa). Furthermore, biological screening for the most active compound, **6b**, showed that it was found to work via more than one mechanism of action, in a “multitarget phenomenon”. Compound **6b** induced apoptosis by increasing the gene expression of Bax, caspase-3, and caspase-9, and the suppression of Bcl-2. Likewise, it caused cell-cycle arrest at the G2/M phase. In addition to its effect on apoptosis and the cell cycle, compound **6b** appeared to inhibit various kinases, such as CDK2, EGFR, HER2, and VEGFR2, which are overexpressed in cancer cells. Thus, it is worthwhile to conclude that the synthesized compounds have a promising anticancer effect via multiple mechanisms, which give these compounds the privilege of avoiding cancer resistance and increasing efficacy.

## Data Availability

Not applicable.

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
