# Peer review of "Design, Synthesis, and Biological Evaluation of 2-Mercaptobenzoxazole Derivatives as Potential Multi-Kinase Inhibitors"

_pharmaceuticals, 2023, doi:10.3390/ph16010097_

Round 1
Reviewer 1 Report
In this manuscript, the authors gave an attractive story about Design, Synthesis and Biological Evaluation of 2-Mercaptoben- 2
zoxazole Derivatives as Potential Multi-kinase Inhibitors. 12 compounds were designed and synthesized based on 2-mercaptobenzoxa- 8
zole and evaluated against several cancer cell lines. It sounds OK for publication unless suggestions below are considered.
1. Please confirm and examine the manuscript by native speaker. Some sentences in abstract and introduction are difficult to follow. Please pay attention to the adjective used in the manuscript, such as line 241, I don’t think the cytotoxicity around 2-6 uM can be called “excellent”.
2. The introduction is too long and should be shortened - a number of the discussion points are now well known to those working in the field and could easily be covered using reference to other literature articles.
3. Why the authors chose the four cell lines for their cytotoxicity evaluation? The correlation of these cells with protein kinases should be demonstrated. The effect of selected compound on protein kinases in cells should also be performed by western blot or other experiments.
4. The western blot is required for evaluation of the level of apoptotic markers in 2.2.5. Furthermore,both the results of 2.2.5. and 2.2.6 of 6b should be showed dose dependently.
Author Response
Reviewer 1
In this manuscript, the authors gave an attractive story about Design, Synthesis and Biological Evaluation of 2-Mercaptoben- 2 zoxazole Derivatives as Potential Multi-kinase Inhibitors. 12 compounds were designed and synthesized based on 2-mercaptobenzoxa- 8 zole and evaluated against several cancer cell lines. It sounds OK for publication unless suggestions below are considered.
Please confirm and examine the manuscript by native speaker. Some sentences in abstract and introduction are difficult to follow. Please pay attention to the adjective used in the manuscript, such as line 241, I don’t think the cytotoxicity around 2-6 uM can be called “excellent”.
R: The manuscript was thoroughly reviewed, and English language was improved.
The introduction is too long and should be shortened - a number of the discussion points are now well known to those working in the field and could easily be covered using reference to other literature articles.
R: The introduction was shortened as suggested.
Why the authors chose the four cell lines for their cytotoxicity evaluation? The correlation of these cells with protein kinases should be demonstrated. The effect of selected compound on protein kinases in cells should also be performed by western blot or other experiments.
R: A panel of four cancer cell lines were used because each cell lines may express different percentage of protein kinases. We designed our compounds with an aim that those compounds may have a multiple kinase activity and we were able to initially prove that. We believe that the best compound of those synthesized hybrids deserves a full investigation of deep mechanistic studies to explore the real mechanism of the antitumor action, therefore, our next research with a pharmacology team will include the suggested comment by the respected reviewer.
The western blot is required for evaluation of the level of apoptotic markers in 2.2.5. Furthermore both the results of 2.2.5. and 2.2.6 of 6b should be showed dose dependently.
R: The apoptotic markers in this study were measured by Enzyme-linked immunosorbent assay (ELISA). ELISA is one of the most specific and straightforward assays for detecting protein levels and therefore it has been reported in many researches as an alternative method for western blot analysis as in the following examples:
Enzyme-Linked Immunosorbent Assay (ELISA). Methods Mol. Biol. 2022, 2508, 115–134, doi:10.1007/978-1-0716-2376-3_10.
Synthesis, Anticancer, Apoptosis-Inducing Activities and EGFR and VEGFR2 Assay Mechanistic Studies of 5,5-Diphenylimidazolidine-2,4-Dione Derivatives: Molecular Docking Studies. Saudi Pharm. J. 2019, 27, 682–693, doi:10.1016/j.jsps.2019.04.003.
A New CDK2 Inhibitor with 3-Hydrazonoindolin-2-One Scaffold Endowed with Anti-Breast Cancer Activity: Design, Synthesis, Biological Evaluation, and In Silico Insights. Mol. 2021, Vol. 26, Page 412 2021, 26, 412, doi:10.3390/MOLECULES26020412.
In addtion, the induction of those markers is frequently done at the IC50 value (fixed dose) of the test compound against the selected cell line. The above references are good examples for that.

Reviewer 2 Report
No comments or suggestions.
Author Response
Reviewer 2
No comments or suggestions

Reviewer 3 Report
This manuscript by Alanazi et al. describes “Design, Synthesis and Biological Evaluation of 2-Mercaptobenzoxazole Derivatives as Potential Multi-kinase Inhibitors. These compounds were tested against hepatocellular carcinoma (HepG2), mammary gland (MCF-7), breast cancer (MDA-MB-231) and epithelioid Cervix Carcinoma (HeLa) cancer cell lines. Compounds were characterized with 1H NMR, 13C NMR, and elemental analyses. Most compounds were obtained as a mixture of E and Z isomers. Some compound displayed good cytotoxic activity and kinase inhibitory activity in comparison to controls. I would recommend this manuscript to be published after as below.
1. Draw chemical structures in correct format in scheme 1
2. The cell line name in table 2 should be MDA-MB-231. Authors should test these compounds against at least one non-cancerous cell lines also.
3. Please correct spelling mistakes (line 174 vales; line 499 clls); check whole manuscript for typographical errors.
4. Figures 2, 3, and 4 are not clearly visible. Increase the resolution of these figures. Authors should provide superimposed structures of reference compounds with different kinases as shown in figures 2-4. Did authors observe similar kind of interaction in active site as reference compounds? Please include this all information in the revised version of manuscript.
5. Please write exact procedure for ELISA assay performed for apoptosis detection. Please also include number of Hep-G2 cells used this assay.
6. Provide flow cytometry analysis image of different phases of cell cycle after treating with compound 6b.
Author Response
Reviewer 3
This manuscript by Alanazi et al. describes “Design, Synthesis and Biological Evaluation of 2-Mercaptobenzoxazole Derivatives as Potential Multi-kinase Inhibitors. These compounds were tested against hepatocellular carcinoma (HepG2), mammary gland (MCF-7), breast cancer (MDA-MB-231) and epithelioid Cervix Carcinoma (HeLa) cancer cell lines. Compounds were characterized with 1H NMR, 13C NMR, and elemental analyses. Most compounds were obtained as a mixture of E and Z isomers. Some compound displayed good cytotoxic activity and kinase inhibitory activity in comparison to controls. I would recommend this manuscript to be published after as below.
- Draw chemical structures in correct format in scheme 1
R: The chemical structure of scheme 1 have been corrected as suggested.
- The cell line name in table 2 should be MDA-MB-231. Authors should test these compounds against at least one non-cancerous cell lines also.
R: The name of MDA-MB-231 has been rewritten as suggested. Regarding the normal cell, this type of cells is not available in our lab so far, however, we are planning to obtain them soon and we will test the cytotoxicity of those compounds against normal cell line when we do our next research on investigating deep mechanistic study of the most active compound of those hybrids.
- Please correct spelling mistakes (line 174 vales; line 499 clls); check whole manuscript for typographical errors.
R: The word has been corrected and the entire manuscript has been thoroughly reviewed for English language correction.
- Figures 2, 3, and 4 are not clearly visible. Increase the resolution of these figures. Authors should provide superimposed structures of reference compounds with different kinases as shown in figures 2-4. Did authors observe similar kind of interaction in active site as reference compounds? Please include this all information in the revised version of manuscript.
R: the resolution of Figures 2,3 and 4 have been modified as suggested. The similarity between the reference compound and compound 6b in binding to the active sites has been discussed as suggested. The figures of superimposition of compound 6b and the reference compounds have been added to the supplementary and discussed in under its related section in the manuscript.
- Please write exact procedure for ELISA assay performed for apoptosis detection. Please also include number of Hep-G2 cells used this assay.
R: The detailed procedures for ELIZA apoptosis detection were written and highlighted. Also, he number of Hep-G2 cells used was written as suggested.
- Provide flow cytometry analysis image of different phases of cell cycle after treating with compound 6b.
R: The image of different phases of the cell cycle after treatment with compound 6b has been added to the manuscript as suggested.

Round 2
Reviewer 1 Report
The author addressed most of my concerns in their revised manuscript. It may be published in present form.
Author Response
The author addressed most of my concerns in their revised manuscript. It may be published in present form.
Thank you.
Reviewer 3 Report
Authors provided response to comments, corrected manuscript, and performed additional studies as per suggestions. My recommendation for current version of manuscript is "Accept in present form" after correcting structures of compound 6a, 6b, and 4a-d in scheme-1. There should not be an extra methylene (-CH2-) in between imine and five-membered heterocycles/phenyl rings (scheme-1).
Author Response
Authors provided response to comments, corrected manuscript, and performed additional studies as per suggestions. My recommendation for current version of manuscript is "Accept in present form" after correcting structures of compound 6a, 6b, and 4a-d in scheme-1. There should not be an extra methylene (-CH2-) in between imine and five-membered heterocycles/phenyl rings (scheme-1).
Response:
Scheme-1 has been corrected as suggested (the extra methylene was removed from compound 6a, 6b, and 4a-d). Thank you.